

# Plant-derived angiogenin fusion protein's cytoprotective effect on trabecular meshwork damage induced by Benzalkonium chloride in mice

Jae Hoon Jeong[1,2,3], Soo Jin Lee[4], Kisung Ko[5], Jeong Hwan Lee[5], Jungmook Lyu[3,6], Moon Hyang Park[7], Jaeku Kang[2,8] and Jae Chan Kim[4]

[1] Department of Ophthalmology, Konyang University Hospital, Daejeon, South Korea
[2] Myunggok Medical Research Institute, Konyang University, Daejeon, South Korea
[3] Myunggok Eye Research Institute, Konyang University, Daejeon, South Korea
[4] Department of Ophthalmology, Chung-Ang University Hospital, Seoul, South Korea
[5] Therapeutic Protein Engineering Lab/College of Medicine, Chung-Ang University, Seoul, South Korea
[6] Department of Medical Science, Konyang University, Daejeon, South Korea
[7] Department of Pathology, Konyang University Hospital, Daejeon, South Korea
[8] Department of Pharmacology/College of Medicine, Konyang University, Daejeon, South Korea

Corresponding author
Jae Chan Kim, jck50ey@daum.net

## ABSTRACT

**Background:** Benzalkonium chloride (BAK), commonly used in glaucoma treatment, is an eye drop preservative with dose-dependent toxicity. Previous studies have observed the multi-functional benefits of angiogenin (ANG) against glaucoma. In our study, we evaluated ANG's cytoprotective effect on the trabecular meshwork (TM) damage induced by BAK. Additionally, we developed a plant-derived ANG fusion protein and evaluated its effect on TM structure and function.

**Methods:** We synthesized plant-derived ANG (ANG-FcK) by fuzing immunoglobulin G's Fc region and KDEL to conventional recombinant human ANG (Rh-ANG) purified from transgenic tobacco plants. We established a mouse model using BAK to look for degenerative changes in the TM, and to evaluate the protective effects of ANG-FcK and Rh-ANG. Intraocular pressure (IOP) was measured for 4 weeks and ultrastructural changes, deposition of fluorescent microbeads, type I and IV collagen, fibronectin, laminin and α-SMA expression were analyzed after the mice were euthanized.

**Results:** TM structural and functional degeneration were induced by 0.1% BAK instillation in mice. ANG co-treatment preserved TM outflow function, which we measured using IOP and a microbead tracer. ANG prevented phenotypic and ultrastructure changes, and that protective effect might be related to the anti-fibrosis mechanism. We observed a similar cytoprotective effect in the BAK-induced degenerative TM mouse model, suggesting that plant-derived ANG-FcK could be a promising glaucoma treatment.

## INTRODUCTION

Glaucoma is a progressive optic neuropathy associated with various risk factors, including increased intraocular pressure (IOP) (*Van Buskirk & Cioffi, 1992*). IOP-related aqueous humor dynamics are currently the only known controllable factors for disease progression prevention. IOP-lowering eye drops contain therapeutic agents and additives (*Inoue, 2014*) such as benzalkonium chloride (BAK), a common ophthalmic preservative agent. Preservatives used in topical eye drops may cause ocular surface disorders, including superficial punctate keratitis, corneal erosion, conjunctival allergy, conjunctival injection and anterior chamber inflammation (*Baudouin, 2008*; *Noecker & Miller, 2011*; *Rosin & Bell, 2013*).

After repeated instillations, BAK penetrates healthy eyes and is detected in both ocular surface structures and deeper tissues, such as the trabecular meshwork (TM) and optic nerve (*Brignole-Baudouin et al., 2012*). Eye drop preservatives may cause long-term trabecular degeneration and increased outflow resistance (*Baudouin et al., 2012*; *Chang et al., 2015*). Chronic or repeated eye drop use can have dose-dependent toxic effects, and should be examined when managing glaucoma. Although there are currently many anti-glaucomatous drugs formulated without preservatives, BAK is still frequently used in medicated eyedrops. Therefore, investigating protective agents against BAK-induced TM damage may improve the treatment and prevention of glaucoma.

Angiogenin (ANG), also known as Ribonuclease 5, has various functions (*Gao & Xu, 2008*) and associations with cancer and neurological diseases via its roles in angiogenesis and apoptosis suppression (*Li, Yu & Hu, 2012*; *Steidinger, Standaert & Yacoubian, 2011*; *Tello-Montoliu, Patel & Lip, 2006*). ANG is highly concentrated in normal tear fluid that has pooled overnight and helps maintain corneal avascularity. It is suggesting that ANG plays a physiological role which is separate from its angiogenic role under normal ocular surface conditions (*Sack et al., 2005*). In addition, it has been reported that ANG could be a candidate survival booster for transformed human TM cell lines (*Kim et al., 2016*). However, the complete effects of BAK at the TM ultrastructure level and ANG's protective mechanism are unclear. Since the trabecular outflow pathway of mice is structurally and functionally similar to that of primates (*Overby et al., 2014*), we examined the protective effect of ANG against BAK in mice TM. Additionally, considering that recombinant human ANG (Rh-ANG) is expensive and plants are beneficial as a heterologous expression system for large scale recombinant protein production (*Jamal et al., 2009*), we developed a plant-derived ANG fusion protein using molecular farming. The aim of our study was to evaluate ANG's efficacy in protecting TM structure and function, and to introduce molecular farming technology to the ophthalmology field.

## MATERIALS AND METHODS

### BAK-induced TM degenerative mouse model

All mouse experiments were conducted in compliance with the Association for Research in Vision and Ophthalmology Statement for the Use of Animals in Ophthalmic and Vision Research. The Institutional Animal Care and Use Committee in College of Medicine,

Konyang University reviewed and approved the study protocol (P-16-22-A-01). The following toxicity model was used: instillation of one drop of 0.01%, 0.02%, 0.1% and 0.2% BAK (Sigma Aldrich, Fluka, Buchs, Switzerland), respectively, twice a day (at 8 AM and 8 PM) for 1 month, and a subconjunctival injection of 10 µL of 0.1% BAK. The mice's contralateral eye served as sham-operated controls. Subconjunctival intramuscular injection of a tiletamine and zolazepam-mixed agent (2 mg/kg, Zoletil; Virbac, Fort Worth, TX, USA) and xylazine (6 mg/kg, Rompun; Bayer, Leverkusen, Germany) was performed under general anesthesia.

Following a previous study where six rats were used to model BAK-induced TM degeneration (Baudouin et al., 2012), we initially assigned five mice to each toxicity model. However, four to five mice were included in each group due to unexpected deaths during the follow-up period. They were all male C57BL/6J Jms SLC mice (7 weeks old; 21–24 g) purchased from SLC Laboratory (Hamamatsu, Shizuoka, Japan). The mice were housed in clear cages with 12-h light/12-h dark cycles at 30–70% humidity and 22–24 °C. Before BAK administration, the mice were allowed to acclimatize for 1 week and were provided tap water and food ad libitum. Their IOP was measured at 6 PM daily using a rebound tonometer (Tono-lab, iCare, Vantaa, Finland) without sedation, and we recorded the average values of three consecutive measurements. After 16 weeks, the mice were euthanized, and their eyes were enucleated for histological analyses. The experimental protocols are summarized in Fig. S1.

## Plant-derived ANG-FcK protein development

We cloned cDNA fragments encoding the human ANG fuzed Fc region of immunoglobulin G-tagged endoplasmic reticulum retention signal, KDEL (ANG-FcK), into a pBI121 plant expression vector. The gene was then inserted with the alfalfa mosaic virus (AMV) untranslated leader sequence from RNA4 under the control of the cauliflower mosaic virus 35S promoter into the vector. We transferred the ANG-FcK gene expression cassette as a HindIII-EcoRI fragment into the plant binary vector pBI121 and conducted Agrobacterium-mediated plant transformation using the vector to generate transgenic tobacco (Nicotiana tabacum) lines expressing ANG-FcK.

We homogenized 100 mg of transgenic plant leaf tissue in 300 µL of 1× PBS, resolved the plant extracts by 12.5% sodium dodecyl sulfate polyacrylamide gel electrophoresis (SDS–PAGE), and transferred them to a nitrocellulose membrane (Millipore, Bedford, MA, USA). The membrane was incubated in blocking solution (5% (w/v) skim milk (Fluka) in 1× TBS, 0.05% (v/v) Tween 20 (TBST)), followed by a primary anti-ANG antibody (1:250, Abcam Inc., Cambridge, MA, USA), and an anti-mouse IgG2a Fc fragment (1:3,000, Jackson ImmunoResearch Labs, West Grove, PA, USA) conjugated to horseradish peroxidase was used as the secondary antibody to detect ANG-FcK. The anti-ANG antibody we used for immunoblotting recognized the full length ANG protein (Cat.# ab10600; Abcam Inc., Cambridge, MA, USA). We used SuperSignal chemiluminescence substrate (Pierce, Rockford, IL, USA) to detect the signal. Rh-ANG (R&D Systems, Minneapolis, MN, USA) was used as a positive control.

We used the same method of purifying plant-derived ANG-FcK as in our previous article (*Lim et al., 2014*). To purify plant-derived ANG-FcK, tobacco leaves were mixed with cold extraction buffer (37.5 mM Tris–HCl pH 7.5, 50 mM NaCl, 15 mM EDTA, 75 mM sodium citrate and 0.2% sodium thiosulfate) and were homogenized in a HR2094 blender (Philips, Seoul, Korea). After homogenization, the leaves were centrifuged for 30 min at $8,800 \times g$ at 4 °C, the supernatant was filtered using Miracloth (Merck, Darmstadt, Germany), and extra pure acetic acid was added to adjust the pH to 5.1. We centrifuged the solution at $10,200 \times g$ for 30 min at 4 °C, brought up the pH to 7.0 by adding 3 M Tris–HCl, and added ammonium sulfate to a saturation of 8%. After centrifugation at $8,800 \times g$ for 30 min at 4 °C, we discarded the precipitate and added ammonium sulfate to the supernatant to 40% saturation. After overnight incubation at 4 °C, the solution was centrifuged, the pellet was resuspended in extraction buffer to 1/10 of the original volume and the final solution was centrifuged at $10,200 \times g$ for 30 min at 4 °C. The supernatant was filtered through a 0.45-mm filter and loaded onto a HiTrap Protein A column (Pharmacia, Uppsala, Sweden). We applied soluble protein extract to a protein A column (GE Healthcare, Piscataway, NJ, USA) and dialyzed elutes of plant-derived ANG-FcK protein against 1× PBS buffer. Aliquots were frozen in liquid nitrogen and stored at −80 °C for glycosylation analysis.

## ANG treatment on the experimental mouse model

We used the 0.1% BAK treatment toxicity model twice daily for 1 month to maximize the toxic effect. Two types of ANG (Rh-ANG and ANG-FcK) were used, and four µL of ANG (50 µg/mL) was administered to mice twice daily for 1 month. We arranged the combinations of toxic and protective substances into six groups: BAK, Rh-ANG, ANG-FcK, Rh-ANG with BAK, ANG-FcK with BAK and sham-treated control. In each experimental group, mice were analyzed using three different methods: three underwent ultrastructural analysis, three underwent immunohistochemical analysis, and three underwent microbead injection to analyze the outflow pathway. Mice were treated with ANG 3 days before BAK administration and the two substances were administered at 10-min intervals.

IOP was measured at 6 PM daily without sedation, and mice were euthanized 4 weeks after BAK and/or ANG treatment. Their eyes were then prepared for electron microscopy or immunohistochemistry. Microbeads were injected into three eyes in each experimental group before mice were sacrificed under general anesthesia to evaluate the conventional outflow pathway. The anterior chambers of eyes were cannulated with a 30-gauge needle connected by tubing to a one-mL syringe filled with green fluorescent beads (100 nm, carboxylate modified FluoSpheres, 1:750 dilution; Molecular Probes, Eugene, OR, USA) and were loaded into a microdialysis infusion pump (World Precision Instruments, Sarasota, FL, USA). A total of 10 µL of liquid was infused into the anterior chamber at 0.167 µL/min for 1 h. The experimental protocols are summarized in Fig. S1.

## Immunohistochemical and ultrastructural analyses

We embedded and froze 36 eyes in Optimal Cutting Temperature Compound (Tissue-Tek, Cat #4583; Sakura Americas, Torrance, CA, USA). Sagittal cryosectioning was performed

through the entire anterior–posterior extension of the globe at a thickness of 10-μm. Sections were stored at −80 °C and dried for 10 min at room temperature. After we washed the sections three times with PBS (Welgene, Gyeongsangbuk-do, Korea) for 10 min each, we drew circles along the tissues using a PAP pen (Vector, Burlingame, CA, USA). The sections were fixed with 4% paraformaldehyde (Santa Cruz Biotechnology, Santa Cruz, CA, USA) for 15 min, incubated with 0.1% Triton X-100 (Sigma–Aldrich, St. Louis, MO, USA) for 5 min for permeabilization, and washed three times with PBS for 10 min each. Slides were incubated in PBS and 1% BSA (Gibco; Thermo Fisher Scientific, Inc., Waltham, MA, USA) for 1 h at room temperature for blocking. Sections were washed once for 10 min and incubated with primary antibodies in blocking solution at 4 °C overnight. Primary antibodies included collagen type I (ab34710, 1:100; Abcam, Cambridge, UK), collagen type IV (ab6586, 1:100; Abcam, Cambridge, UK), fibronectin (sc-69681, 1:100; Santa Cruz), laminin (ab11575, 1:100; Abcam, Cambridge, UK), and α-smooth muscle actin (α-SMA) (sc-53142, 1:100; Santa Cruz). After being washed three times (10 min each), sections were incubated for 1 h with Cy2 (green) and Cy3 (red) secondary antibodies (1:250; Jackson ImmunoResearch, West Grove, PA, USA). Sections were washed, counter-stained with Hoechst 33258 (1:1,000), and mounted with a drop of AquaPolyMount (Polysciences, Warrington, PA, USA). We obtained images using the fluorescence microscope, Imager D2 (Zeiss, Oberkochen, Germany).

For ultramicroscopy, we fixed 18 eyes overnight in cold 2.5% glutaraldehyde, then in one osmium tetroxide for 1 h. After dehydration in a graded acetone series, tissues were embedded in Epon resin, and 0.5-μm semithin or 70-nm ultrathin sections were cut using an ultramicrotome. Semithin sections were stained with toluidine blue. Ultrathin sections were placed on 200-mesh copper grids and double stained with 4% uranyl acetate for 20 min and 0.2% lead citrate for 5 min. To obtain tangential sections parallel to the plane of the inner wall, we took consecutive semithin sections through the cornea and sclera in a plane parallel to the limbus's outer surface. Once we reached the lumen of Schlemm's canal (SC), we removed consecutive ultrathin sections until we reached the inner wall of SC, juxtacanalicular connective tissue (JCT), and the lamellated TM. Semithin sections were viewed using an Olympus CX22 microscope (Tokyo, Japan), and ultrathin sections were viewed using an electron microscope (HT7700; Hitachi High-Tech Science Corp., Tokyo, Japan) at 80 kV.

## Statistical analysis

Our results are expressed as means ± standard errors, and normality and equal variances in groups were tested. Analysis of variance (ANOVA) was used to analyze IOP differences across three or more groups at each timepoint, and repeated measures ANOVA was used to compare the baseline IOP during the follow-up period. We included Tukey's tests, Bonferroni's methods, Duncan's tests, and Dunnett's T3 tests in post-hoc analyses. The probability level for statistical significance was set at 5%. Data were recorded and analyzed using SPSS for Windows, version 18.0 (SPSS Inc., Chicago, IL, USA).

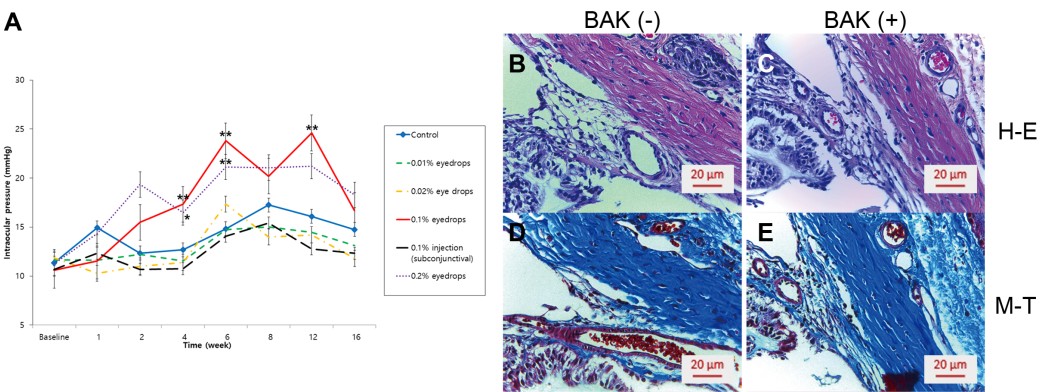

**Figure 1 Changes in intraocular pressure (IOP) in response to BAK in mouse models and representative histological stains of the mouse eye TM.** (A) The mean IOP increased significantly in response to the instillation of 0.1% and 0.2% BAK at 4 weeks, and remained higher than that of the control group at 6 weeks, although the instillation was maintained for 4 weeks. Error bars represent standard errors of the mean. $^*P < 0.05$ and $^{**}P < 0.01$ (compared to the control group). (B) Hematoxylin and eosin (H–E) stain of the mouse eye TM under a light microscope. (C) H–E stain of the characteristic architecture of outflow tissues showed little difference by BAK. (D) Masson trichrome (M–T) stain of the mouse eye TM under a light microscope. (E) M–T stain of the characteristic architecture of outflow tissues showed little difference 16 weeks after BAK administration.

## RESULTS

### BAK effect on intraocular pressure

The mean IOP changes in response to various BAK concentrations are shown in Fig. 1A. After 2 weeks of treatment twice a day, 0.1% and 0.2% BAK increased the mean IOP. The IOP was higher in these groups than in the sham-operated control group over the 16-week period, although treatment was performed for 4 weeks. At 4 weeks, 0.1% BAK treatment significantly induced an increase in IOP by approximately 36% (17.3 ± 1.0 mmHg) compared to that of the control group (12.7 ± 0.6 mmHg, $P < 0.01$). This group's IOP remained higher at 6 weeks (23.8 ± 1.2 mm Hg) and 12 weeks (24.6 ± 2.3 mmHg) than the control group at the same points in time (14.8 ± 3.0 mmHg and 16.1 ± 5.1 mmHg, $P < 0.01$).

The mean IOP was significantly higher in the 0.2% BAK group than in the control group at 4 weeks (16.5 ± 2.4 mmHg, $P = 0.02$) and at 6 weeks (21.1 ± 1.3 mmHg, $P < 0.01$). The IOP under 0.01% and 0.02% BAK treatments and subconjunctival 0.1% BAK injection was not significantly different from that of the control group, except for 0.02% BAK at 1 week (10.3 ± 1.3 mmHg vs. 14.9 ± 1.6 mmHg in the control group; $P = 0.03$). In the toxic BAK-induced TM degeneration group, we administered a 0.1% BAK treatment twice a day for 4 weeks.

### Expression and purification of ANG-FcK in transgenic plants

We examined ANG-FcK expression in randomly selected transgenic plants using western blotting (Fig. 2A). The Rh-ANG protein band was detected at approximately 15 kDa and

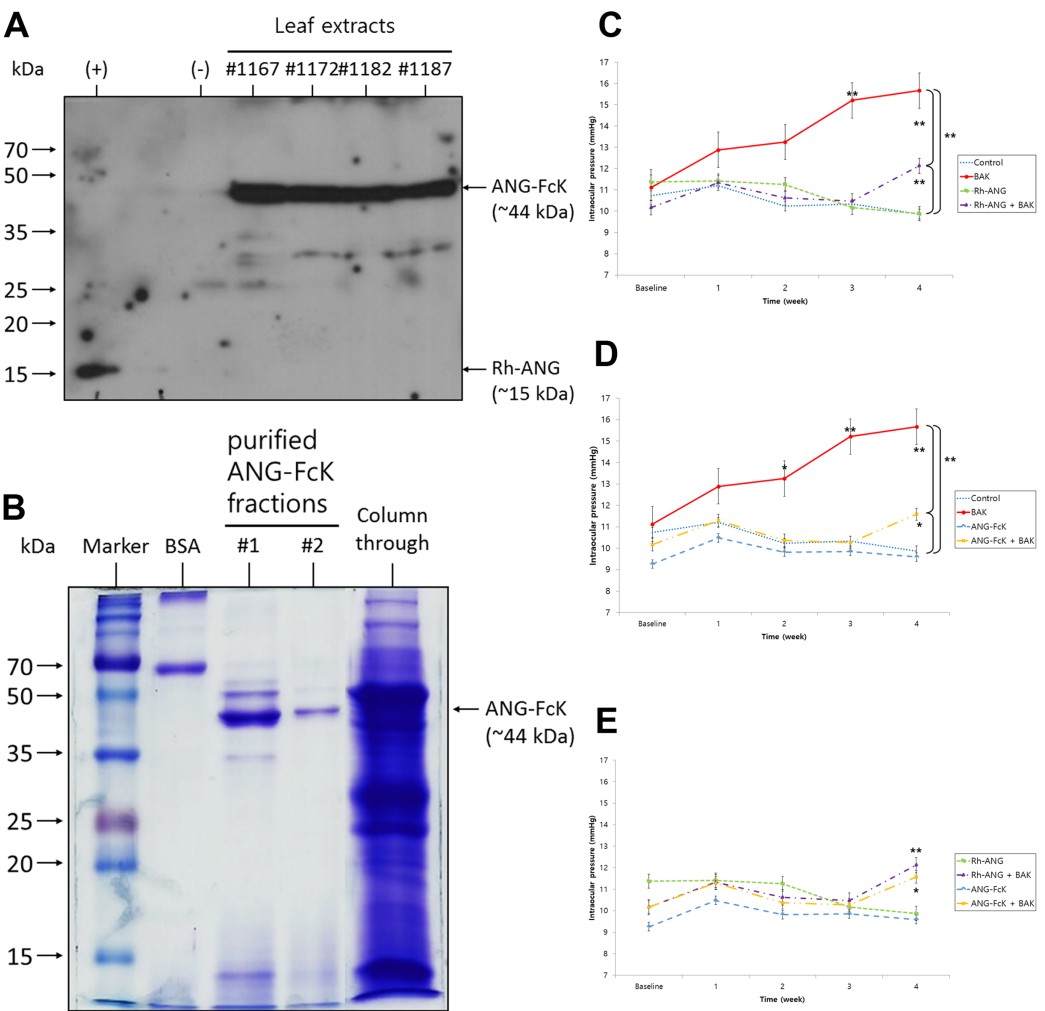

**Figure 2** **Development of ANG-FcK, and changes in the effects of BAK on intraocular pressure (IOP) in response to ANG.** (A) Expression of ANG-FcK in randomly selected transgenic plants. (+), positive control, Rh-ANG; (−), non-transgenic tobacco plant leaf extract. #1167–1187, transgenic plant line number. (B) SDS–PAGE results for purified ANG-FcK. #1–2, purified protein fraction number; Column through: plant extracts passed through a column. (C) The single BAK treatment induced the greatest elevation in mean IOP after 3 weeks among all groups. Cotreatment with Rh-ANG and BAK maintained the initial mean IOP over 3 weeks; however, IOP was elevated at 4 weeks, although it was lower than that for the single BAK treatment group. (D) The change in mean IOP for co-treatment with ANG-FcK and BAK was similar to that for Rh-ANG. IOP was higher than that in the control group, but lower than that in the single BAK treatment group at 4 weeks. (E) There was no significant difference in mean IOP between Rh-ANG and ANG-FcK for the combined use with BAK. Error bars represent standard errors of the mean. $^*P < 0.05$ and $^{**}P < 0.01$.

ANG-FcK was detected at approximately 44 kDa. No band was observed in the non-transgenic plant. We purified ANG-FcK from leaves harvested from transgenic tobacco plants. Protein A column purification yielded an average of 2 mg of plant-derived ANG-FcK per kg of fresh leaves from a line with high protein expression. SDS–PAGE analysis of purified ANG-FcK revealed one major band (44 kDa, Fig. 2B).

## Effect of ANG on BAK-induced changes in intraocular pressure

The mean IOP after 0.1% BAK treatment continued to increase and was significantly higher than that of the other groups at 3 weeks ($15.2 \pm 2.1$ mmHg, $P < 0.01$) and 4 weeks ($15.7 \pm 1.7$ mmHg, $P < 0.01$) (Figs. 2C and 2D). There were few IOP differences between the single Rh-ANG or ANG-FcK treatments and the control group, and inter- and intra-group variability was low (Fig. 2E). For treatments with Rh-ANG or ANG-FcK with BAK, the mean IOP was similar to those of single Rh-ANG and ANG-FcK treatments and the control group at 3 weeks. However at 4 weeks, Rh-ANG with BAK was elevated to $12.1 \pm 1.8$ mmHg ($P < 0.01$) and ANG-FcK with BAK to $11.6 \pm 0.4$ mmHg ($P < 0.05$), although these values were lower for the single BAK group ($P < 0.01$) (Figs. 2C–2E).

## Immunohistochemical analysis of the effects of ANG on BAK response

We observed that type I collagen's Cy3 labeling in the outflow tissue along the iridocorneal angle was more pronounced in BAK-treated eyes than in the single ANG and control groups (Fig. 3A). The type Ⅰ collagen labeling in the TM region adjacent to the corneal endothelium was patchy and thin in the control (Fig. 3A) and single Rh-ANG (Fig. 3B) and ANG-FcK (Fig. 3C) groups. However, we observed more intense and broader labeling in the single BAK (Fig. 3D) and Rh-ANG with BAK (Fig. 3E) and ANG-FcK with BAK (Fig. 3F) groups. This labeling pattern was similar across type IV collagen (Figs. 3G–3L) and fibronectin (Figs. 3M–3R). Type IV collagen and fibronectin labeling was more prominent in BAK-treated eyes (especially those in the single BAK-treated group) than in the single ANG and control groups. The positive laminin labeling was more noticeable in the single BAK-treated group (Fig. 4D) than in the Rh-ANG with BAK (Fig. 4E) and ANG-FcK with BAK (Fig. 4F) groups, but was barely detectable in the single ANG and control groups (Figs. 4A–4C). We only detected spotty positive α-SMA labeling in the inner sclera of the TM layers adjacent to SC in the single BAK-treated group (Fig. 4J), not in any ANG-treated and control groups. In the ciliary body, type I, IV collagen and fibronectin labeling was observed in the epithelium and laminin and α-SMA labeling was more apparent in the stroma layer (Figs. 3 and 4).

Fluorescent bead deposition traces the flow of aqueous humor, and its intensity is correlated with TM outflow function (*Li et al., 2016*; *Swaminathan et al., 2013*). Green fluorescent beads were present in all of the experimental groups' outflow tissues, but deposition intensity decreased in the single BAK-treated group (Fig. 4P). The sparsely deposited fluorescent beads suggest an abnormal TM outflow function resulting from BAK treatment. The cumulative bead distribution data from the single BAK-treated group can be found in Fig. S2. The Rh-ANG and ANG-FcK with BAK groups exhibited more prominent deposition of green fluorescent beads in their outflow tissues, but these results were based on a qualitative analysis.

## TM histological and ultrastructural changes

After BAK treatment, the characteristic structure of outflow tissues showed little difference under a light microscope (Figs. 1B–1E). Based on our observations of the ultrathin sections, BAK treatment led to a thickening of the lamina beam in the TM. In particular,

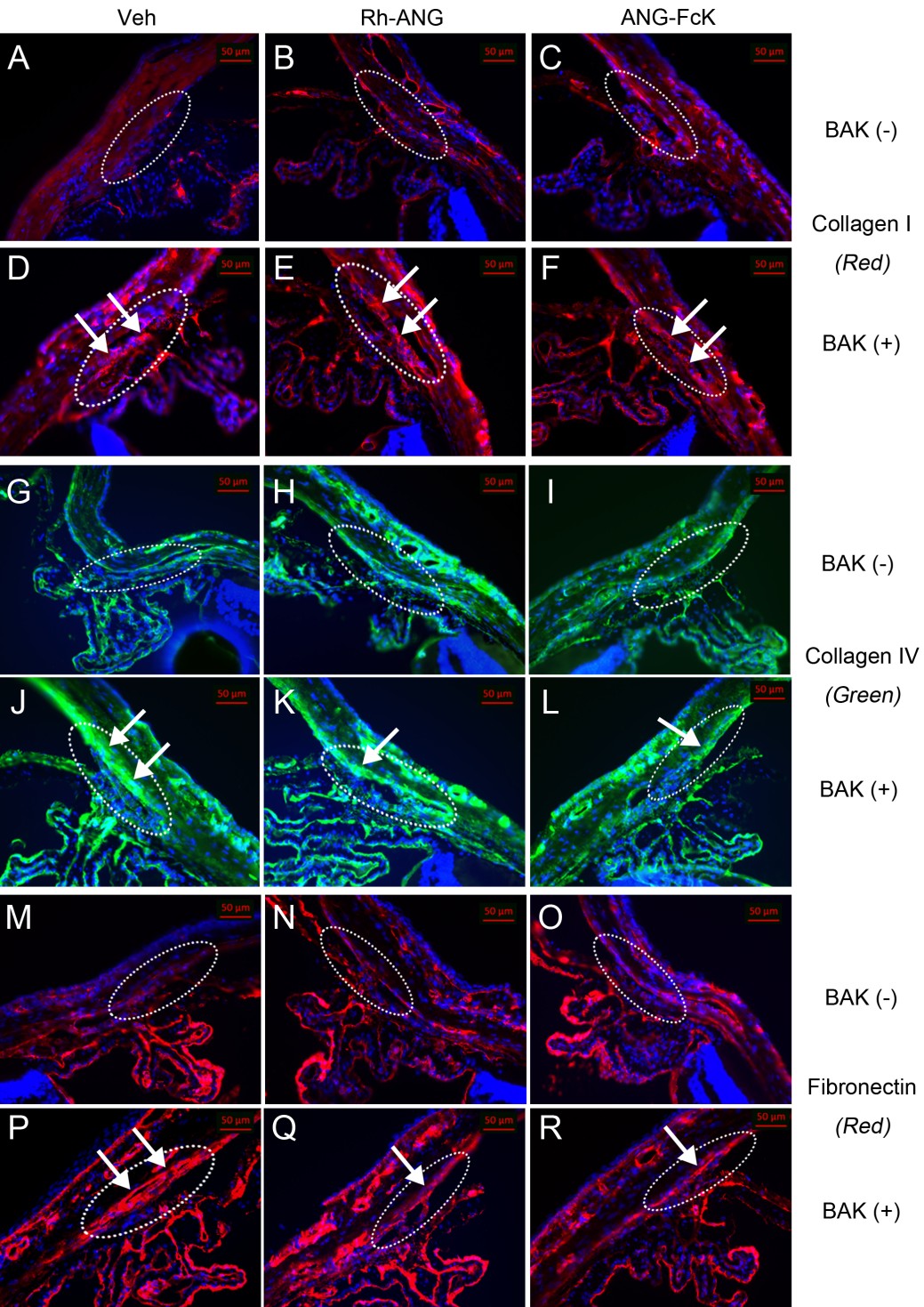

**Figure 3 The first of the two immunohistochemical labeling of aqueous outflow tissues in a mouse model.** (A) Sham-treated control. (B) Single Rh-ANG treatment. (C) Single ANG-FcK treatment. (D) Single BAK treatment. (E) Rh-ANG with BAK treatment. (F) ANG-FcK with BAK treatment group. Staining type I collagen (red) was present in outflow tissues (dotted oval) of the TM and the inner and outer walls of Schlemm's canal in sham-treated control and ANG-treated mice, but the distribution tended to be patchy and thin. BAK-treated mice, in contrast, showed more intense and broader labeling in inner walls of Schlemm's canal (arrows). Nuclei were counter-stained with Hoechst (blue). (G) Sham-treated control.

**Figure 3** (continued)
(H) Single Rh-ANG treatment. (I) Single ANG-FcK treatment. (J) Single BAK treatment. (K) Rh-ANG with BAK treatment. (L) ANG-FcK with BAK treatment group. Staining of type IV collagen (green) was more prominent in BAK-treated mice (arrows) and especially in eyes of the single BAK-treated group than in single ANG and sham-treated control groups. (M) Sham-treated control. (N) Single Rh-ANG treatment. (O) Single ANG-FcK treatment. (P) Single BAK treatment. (Q) Rh-ANG with BAK treatment. (R) ANG-FcK with BAK treatment group. Staining of Fibronectin (red) was more prominent in BAK-treated mice (arrows) and especially in eyes of the single BAK-treated group than in single ANG and sham-treated control groups. All images were magnified 200 times. Veh, vehicle for sham-treated control.

we observed hyperthrophied JCT and the accumulation of fibrillar material within the JCT underlying the SC's inner wall. These were prominent when compared with the empty spaces within the control group's JCT (Figs. 5A and 5B). An increased density and multidirectional array of fibrillar material were also seen in the single BAK-treated group (Figs. 5C and 5D), similar to the "fingerprint"-like basement membrane described in human eyes treated with corticosteroids (Johnson et al., 1997).

There were no remarkable differences between the TM ultrastructures of the Rh-ANG with BAK group (Figs. 5E and 5F) and the ANG-FcK with BAK group (Figs. 5G and 5H). These groups showed some similarities such as a thickness of lamina beam and empty spaces within the JCT. The fibrillar material was denser than that of the control group, but sparser than that of the single BAK-treated group. Although we did not do a quantitative analysis, we found more intracellular organelles in the ANG and BAK-treated groups than in the control group.

## DISCUSSION

In this study, we examined the toxicity of chronic BAK exposure on the TM and ANG's defenses against changes in the trabecular outflow pathway. We induced the structural and functional degeneration of the TM through BAK treatment in a mouse model. Co-treatment with ANG successfully preserved the outflow function of the TM, suggesting that ANG prevents fibrosis.

Additionally, we developed ANG-FcK, which has important practical applications. The greater molecular weight of ANG-FcK (44 kDa) over Rh-ANG (15 kDa) enhanced protein stability, facilitated purification, and improved yield. ANG-FcK was similar to Rh-ANG with respect to IOP, flow of aqueous humor and ultrastructural changes in the BAK-induced TM degenerative mouse model. Purifying plant-derived ANG-FcK yielded an average of 2 mg per kg of fresh leaves. With a cost of approximately U.S. 1,000 dollars per 250 μg of Rh-ANG, 1 kg of transgenic plants is worth approximately U.S. 8,000 dollars of conventional protein. To the best of our knowledge, this is the first study to apply molecular farming techniques in ophthalmology, and our production of recombinant ANG may be beneficial to this field.

Although BAK is the most common preservative used in ophthalmic solutions, its effects on IOP or outflow in vivo have not been explored (Rasmussen, Kaufman & Kiland, 2014). After 0.1% and 0.2% BAK topical drops delivered twice a day for 4 weeks in our mouse model, the IOP rose significantly at 4 weeks and remained elevated for

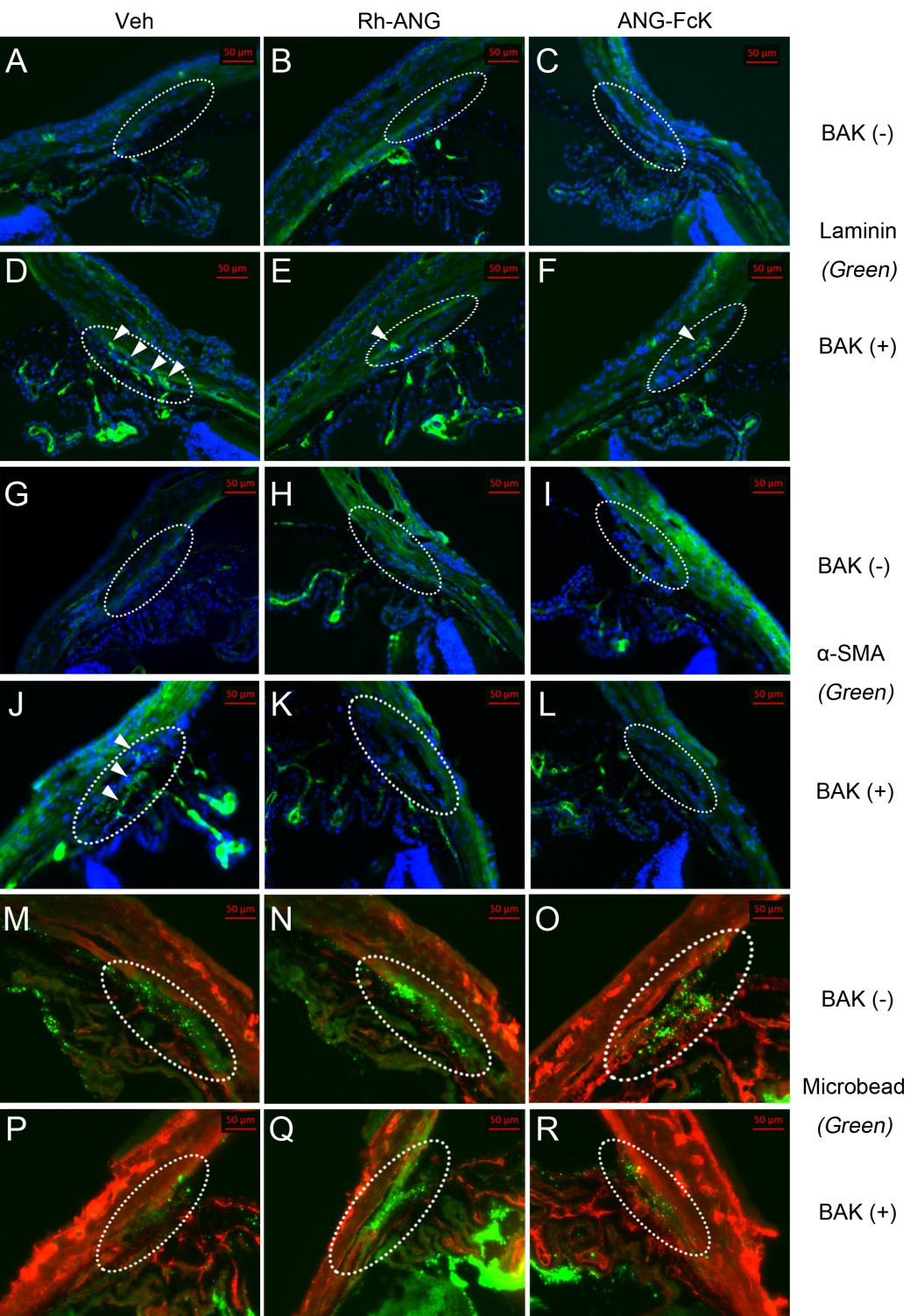

**Figure 4 The second of the two immunohistochemical labeling of aqueous outflow tissues in a mouse model.** (A) Sham-treated control. (B) Single Rh-ANG treatment. (C) Single ANG-FcK treatment. (D) Single BAK treatment. (E) Rh-ANG with BAK treatment. (F) ANG-FcK with BAK treatment group. Staining of laminin (green) in the single BAK-treated group was more pronounced in outflow tissues (dotted oval) of the TM and the inner and outer walls of Schlemm's canal than in the Rh-ANG or ANG-FcK with BAK groups (arrowheads) but it was barely detectable in single ANG and control groups. (G) Sham-treated

**Figure 4 (continued)**
control. (H) Single Rh-ANG treatment. (I) Single ANG-FcK treatment. (J) Single BAK treatment. (K) Rh-ANG with BAK treatment. (L) ANG-FcK with BAK treatment group. Spotty positive α-SMA labeling (green) in the TM layers adjacent to Schlemm's canal (arrowheads) was only detected in the eye of the single BAK-treated group, but not in any ANG-treated and control groups. (M) Sham-treated control. (N) Single Rh-ANG treatment. (O) Single ANG-FcK treatment. (P) Single BAK treatment. (Q) Rh-ANG with BAK treatment. (R) ANG-FcK with BAK treatment group. Green fluorescent beads were deposited in outflow tissues; however, they were sparse in the single BAK-treated group. Type I collagen was used for counter-staining (red). All images were magnified 200 times. Veh, vehicle for sham-treated control.

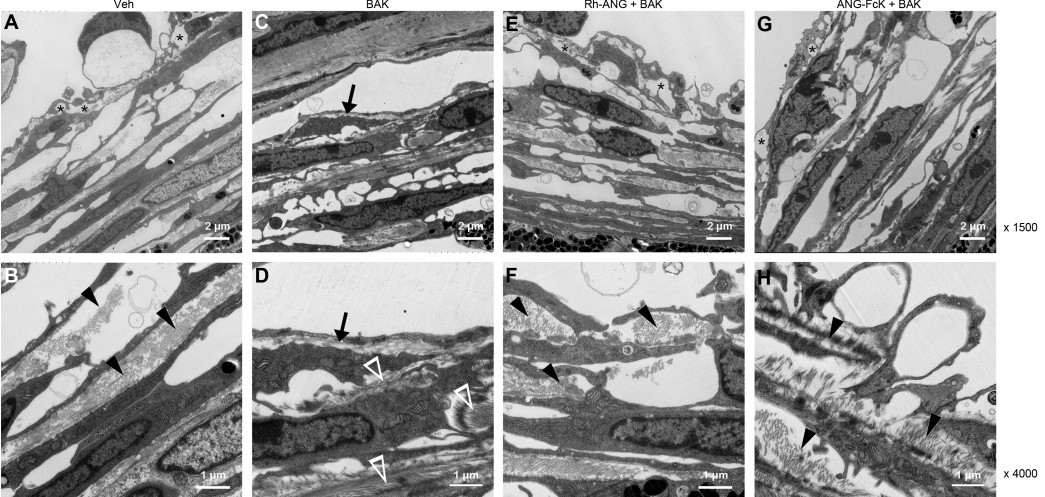

**Figure 5 Ultrastructural changes in the TM of mice treated with the combination of BAK and ANG for 4 weeks.** Photographs with magnification power of 1,500 were in top row, and those with magnification power of 4,000 were in bottom row. (A) In sham-treated control mice without BAK, optically open spaces (asterisks) were often observed between juxtacanalicular connective tissue (JCT) cells. (B) Accumulation of fibrillar material within JCT was sparse and unidirectional (arrowheads). (C) In BAK-treated mice, the thickness of the lamina beam increased within the JCT (arrows). (D) The JCT was often filled with fine fibrillary material that showed an increased density and multidirectional array (empty arrowheads). Increased intracellular organelles were found in the cytoplasm of trabeculocytes with BAK treatment. (E) The ultrastructures of TM for the combination of Rh-ANG with BAK. The thickness of the lamina beam and the empty spaces within the JCT (asterisks) were preserved. (F) Fibrillar materials were denser in the combined treatment group than the control group, but sparser than in the single BAK treatment group (arrowheads). (G) The ultrastructures of TM. (H) Deposition of fibrillar materials for the combination of ANG-FcK with BAK were similar with those of Rh-ANG with BAK.

2 additional weeks. This was higher than the IOP for other concentrations and the 0.1% BAK subconjunctival injection. These findings differ from the results of *Baudouin et al. (2012)* who found that IOP significantly increased 7 days after 100 μL of 0.01% BAK subconjunctival injection and remained high for 6 additional days after a second injection on day 7. A major reason for these differences could be the animal model used. *Baudouin et al. (2012)* injected 100 μL of BAK into the subconjunctival space of rats weighing 300–350 g. In our study, we performed a single injection of 10 μL into the subconjunctival space of mice weighing 21–24 g. This might explain the lack of a significant change in IOP

despite a higher BAK concentration. In a mouse model, it is difficult to inject 10 µL into the subconjunctival space without any losses. Moreover, the administration of topical drops is more suitable for a chronic exposure model.

The IOP of the combination of ANG and BAK was lower than that the IOP of BAK alone, but was greater than that of eyes not exposed to BAK at 4 weeks. Our results are in agreement with the findings of an earlier experimental study (*Kim et al., 2016*) where ANG lowered IOP in both normal and elevated rat models using the vortex vein cauterization method. ANG also conserved the conventional outflow of aqueous humor via the TM after BAK treatment. Fluorescent beads were deposited along the outflow tract in the ANG and control groups, but were sparse in the BAK-induced toxicity model. The cumulative distribution of microbeads was sparse across the parallel sections of all three single BAK-treated mouse models (Fig. S2). Although it was based on a subjective analysis since there are low and high-flow regions in the 360-degree circumference of the TM, these results indicate that BAK creates an abnormal outflow of aqueous humor, which is consistent with previous results (*Swaminathan et al., 2013*; *Zhang et al., 2009*). Cross-sectional images cannot represent the whole TM, but analyses of fluorescence intensities on flat-mounted sections have been suggested as helpful for this task.

Both the lamina thickness and fibrillar material density increased, and we observed type I, type IV collagen, fibronectin, laminin and α-SMA fibrogenic markers (*Faralli, Filla & Peters, 2019*; *Ko & Tan, 2013*; *Pattabiraman, Maddala & Rao, 2014*) in the BAK-induced toxicity model. BAK caused an epithelial mesenchymal transition-like phenomenon and myofibroblast-like phenotypic changes in the TM. These changes caused TM cells to abundantly express fibronectin, activate motility, and switch to a myofibroblast-like phenotype, simultaneously strengthening the actin cytoskeleton and extracellular matrix. Fibronectin regulates the deposition of collagen IV and laminin (*Faralli, Filla & Peters, 2019*). Overall, these changes cause an increase in TM resistance to aqueous humor outflow (*Takahashi et al., 2014*; *Tamm, 2013*). In eyes co-treated with ANG and BAK, we did not detect α-SMA, and the overall ultrastructural configuration was similar to that of the control. However, the fibrillar material density had increased and extracellular matrix markers such as collagen IV, fibronectin and laminin were more abundant than in the single ANG and control groups. These findings confirm that ANG defends against fibrosis and myofibroblast-like phenotypic changes induced by BAK by maintaining the proper ultrastructure for aqueous humor outflow.

Our study has its limitations. First, the BAK concentration we used was higher than those used in commercial eye drops. However, we used BAK that had accumulated in the TM, iris and lens samples during cataract and glaucoma surgery in patients after long-term administration of BAK-containing medication (*Desbenoit et al., 2013*). Second, the immunohistochemical and ultrastructural findings in our study were based on subjective analyses. Quantifiable methods such as measuring effective filtration areas on anterior segment images (*Li et al., 2016*; *Swaminathan et al., 2013*) and ultrastructural analysis of basement membrane material length (*Overby et al., 2014*) are needed for more comprehensive data. Third, extra in vitro experimental studies are necessary to investigate the protective mechanism of ANG against BAK. Previous studies showed that ANG

may activate Akt-mediated signals for nitric oxide production and TM remodeling by regulating matrix metalloproteinase and rho-kinase (*Kim et al., 2016*). Finally, further research on ANG's effect on retinal ganglion cells may clarify its function and improve its clinical effectiveness. Because glaucoma is an ocular neurodegenerative disease characterized by the progressive death of retinal ganglion cells, the importance of ANG enrichment in normal motor neurons has been observed in studies on amyotrophic lateral sclerosis, a fetal neurodegenerative disease (*Kieran et al., 2008*).

## CONCLUSIONS

In conclusion, ANG's protective effect on TM may involve an anti-fibrotic function with a less extensive ultrastructural change that retains outflow function than exposure to single toxic substance such as BAK. Plant-derived ANG-FcK's protective effect is similar to that of Rh-ANG, and it is a promising candidate for an alternative eye drop additive. Future studies should focus on ANG's detailed defense mechanism and potential applications in glaucoma management.

## ACKNOWLEDGEMENTS

The authors wish to thank medical laboratory technologist Dae Young Kim, Department of Pathology, Konyang University Hospital, Daejeon, Korea, for their electron microscopy technical support.

### Funding

This study was supported by the Konyang University Myunggok Research Fund of 2016. The funders had no role in study design, data collection and analysis, decision to publish, or preparation of the manuscript.

### Grant Disclosures

The following grant information was disclosed by the authors:
Konyang University Myunggok Research Fund.

### Competing Interests

The authors declare that they have no competing interests.

### Author Contributions

- Jae Hoon Jeong conceived and designed the experiments, performed the experiments, analyzed the data, prepared figures and/or tables, authored or reviewed drafts of the paper, and approved the final draft.
- Soo Jin Lee conceived and designed the experiments, performed the experiments, prepared figures and/or tables, and approved the final draft.
- Kisung Ko performed the experiments, prepared figures and/or tables, authored or reviewed drafts of the paper, and approved the final draft.

- Jeong Hwan Lee performed the experiments, prepared figures and/or tables, authored or reviewed drafts of the paper, and approved the final draft.
- Jungmook Lyu performed the experiments, authored or reviewed drafts of the paper, and approved the final draft.
- Moon Hyang Park performed the experiments, prepared figures and/or tables, and approved the final draft.
- Jaeku Kang performed the experiments, prepared figures and/or tables, and approved the final draft.
- Jae Chan Kim conceived and designed the experiments, analyzed the data, authored or reviewed drafts of the paper, and approved the final draft.

## Animal Ethics

The following information was supplied relating to ethical approvals (i.e., approving body and any reference numbers):

Institutional Animal Care and Use Committee in College of Medicine, Konyang University provided full approval for this research (16-22-A-01).

## Data Availability

The raw measurements are available in the Supplemental Files.

## Supplemental Information

Supplemental information for this article can be found online at http://dx.doi.org/10.7717/peerj.9084#supplemental-information.

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
