# Peer review of "Plant-derived angiogenin fusion protein’s cytoprotective effect on trabecular meshwork damage induced by Benzalkonium chloride in mice"

_PeerJ, doi:10.7717/peerj.9084_

## Round 0.1 · original submission · Major Revisions

As you can see, one reviewer was happy with your paper but nevertheless made a series of useful comments and suggestions. The other reviewer was less convinced and suggested additional work. If you ready to meet these criticisms, I may consider a revised version of this submission. Please, attach also a rebuttal where you explain, pint by point, how and where in the revised version you made the requested changes and additions. If you disagree with some of the comments, please explain why. Note that your revise version may undergo anew round of review by the same or by different reviewers. I cannot, therefore, make any commitment at this stage about a final acceptance of your work.

Reviewer 1 ·

Basic reporting

Poor English was used.

Experimental design

Authors need to plan some more experiments (mentioned in comments to the author) to full fill their statements

Validity of the findings

The authors over-interpreted their results

Additional comments

Abstract: Background: Adding one sentence about “ANG cytoprotective activity” which is previously published would connect in a better way.

Authors need to check the sentences carefully for ex. “The cytoprotective effect of ANG on TM resulted from a reduction in ultrastructural changes and retained outflow function” . The authors have written the paper with poor English. Need to improve.

Authors did not mention the purpose of ANG fusion protein (ANG-FcK) over the Rh-ANG.

Insert scale in the images

Fig.3A&B: It seems ANG or ANG-FcK did not show any prominent anti-fibrotic effects. Any prominent reduction in Col-I staining is observed up on ANG/ANG-FcK treatment in BAK treated TM tissues. Authors encouraged to examine more ECM markers in these tissues like fibronectin, Col-IV, laminin, elastin etc which help to give a broad statement like anti-fibrotic effect. Even alpha-SMA staining not convincing to the authors statement.

Fig.3C: The low number of green florescent beads were observed in the BAK treated group compared to the control group. This method (outflow) little bit subjective as there are low and high flow regions in 360-degree circumference of the TM. It might possible that the low number of beads in BAK treated group might be due to the plane of the TM section the authors observed. Did authors observe the bead distribution in continuous parallel sections? If so giving the cumulative bead distribution data, more appropriate in this case or observing the distribution of the beads in TM flat whole mounts will give a better idea.

Reviewer 2 ·

Basic reporting

1. The supplemental figure did not download properly or it was formatted wrong because it appeared dark and greyed and was difficult to read.
2. Figure 1B, label the treatment and stain in the figure.
3. Figure 2B, each graph should be a separate B, C, D and referenced so in the text of the Results section for clarity.
4. Figure 3, label which antibody was used for each panel in the figure.
5. Starting at line 263, it would be helpful to introduce the experiment of why beads are being injected into the eye first. So, I would suggest switching the sentence starting at Line 265 with the sentence starting at line 263.

Experimental design

How were the statistics done for the IOP data? Was an ANOVA calculated at each timepoint?

Validity of the findings

Conclusions are well stated.

Additional comments

Jeong et al evaluated the cytoprotective effect of ANG on TM damage induced by BAK. Overall the manuscript was well written, and experiments performed appropriately.

---

## Round 0.2 · accepted · Accept

Both reviewers were happy with your revision. Congratulations.

Reviewer 1 ·

Basic reporting

No comment

Experimental design

No comment

Validity of the findings

no comment

Additional comments

None

Reviewer 2 ·

Basic reporting

Pass

Experimental design

Pass

Validity of the findings

Pass

Additional comments

The authors have addressed all of my concerns.